# Antiasthmatic prescriptions in children with and without congenital anomalies: a population-based study

Natalie Divin [1], Joanne Emma Given [1], Joachim Tan,[2] Gianni Astolfi,[3] Elisa Ballardini,[4] Laia Barrachina-Bonet,[5] Clara Cavero-Carbonell [5], Alessio Coi,[6] Ester Garne [7], Mika Gissler,[8] Anna Heino,[8] Susan Jordan [9], Anna Pierini,[10] Ieuan Scanlon,[9] Stine Kjær Urhøj,[11] Joan K Morris [2], Maria Loane [1]

For numbered affiliations see end of article.

**Correspondence to**
Dr Maria Loane;
ma.loane@ulster.ac.uk

## ABSTRACT

**Objectives** To explore the risk of being prescribed/dispensed medications for respiratory symptoms and breathing difficulties in children with and without congenital anomalies.

**Design** A EUROlinkCAT population-based data linkage cohort study. Data on children with and without congenital anomalies were linked to prescription databases to identify children who did/did not receive antiasthmatic prescriptions. Data were analysed by age, European region, class of antiasthmatic, anomaly, sex, gestational age and birth cohort.

**Setting** Children born 2000–2014 in six regions within five European countries.

**Participants** 60 662 children with congenital anomalies and 1 722 912 reference children up to age 10 years.

**Primary outcome measure** Relative risks (RR) of >1 antiasthmatic prescription in a year, identified using Anatomical Therapeutic Chemical classification codes beginning with R03.

**Results** There were significant differences in the prescribing of antiasthmatics in the six regions. Children with congenital anomalies had a significantly higher risk of being prescribed antiasthmatics (RR 1.41, 95% CI 1.35 to 1.48) compared with reference children. The increased risk was consistent across all regions and all age groups. Children with congenital anomalies were more likely to be prescribed beta-2 agonists (RR 1.71, 95% CI 1.60 to 1.83) and inhaled corticosteroids (RR 1.74, 95% CI 1.61 to 1.87). Children with oesophageal atresia, genetic syndromes and chromosomal anomalies had over twice the risk of being prescribed antiasthmatics compared with reference children. Children with congenital anomalies born <32 weeks gestational age were over twice as likely to be prescribed antiasthmatics than those born at term (RR 2.20, 95% CI 2.10 to 2.30).

**Conclusion** This study documents the additional burden of respiratory symptoms and breathing difficulties for children with congenital anomalies, particularly those born preterm, compared with children without congenital anomalies in the first 10 years of life. These findings are beneficial to clinicians and healthcare providers as they identify children with greater morbidity associated with respiratory symptoms, as indicated by antiasthmatic prescriptions.

## STRENGTHS AND LIMITATIONS OF THIS STUDY

⇒ The use of population-based data from six regions across Europe yielded a large sample size of children with and without congenital anomalies.

⇒ Use of a common data model enabled standardisation of prescription/pharmacy dispensing records across a range of coding classification systems, languages and healthcare systems in Europe.

⇒ Risks of being prescribed/dispensed medications for respiratory conditions were estimated in children with any major congenital anomaly, in children with 32 specified isolated congenital anomalies, and for within region comparisons in children with no major congenital anomaly.

⇒ Over 95% of children across registries were linked to prescription databases limiting the potential bias from missed linkages.

⇒ The lack of information on socioeconomic status meant this potential confounding factor could not be adjusted for.

## INTRODUCTION

Breathing difficulties in childhood are a leading cause of emergency department visits, hospitalisations and missed days at school.[1] The prevalence of chronic breathing conditions such as asthma varies globally in children. The US National Health Interview Survey in 2013 reported prevalence rates of 8.3% in children aged 0–17 years based on parent reports of doctor-diagnosed asthma.[2] The International Study of Asthma and Allergies in Childhood (ISAAC) 1999–2004, which was also based on parental reports, found that prevalence of asthma in European children aged 6–7 years varied from <5% in Albania to over 20% in the UK, with prevalence rates increasing yearly.[3 4] Similarly, the more recent Mechanisms of the Development of ALLergy study involving eight birth cohorts across Europe found that parental reports of

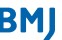

doctor-diagnosed asthma at age 4 years ranged from 1.7% in Germany to 13.5% in Bradford, England.[5]

The prevalence of asthma also varies according to age and sex in childhood. The ISAAC study consistently found higher asthma rates for 13–14 years compared with 6–7 years, and this pattern of higher asthma prevalence rates in older children has been replicated across Europe.[5–7] However, as asthma can be challenging to diagnose in infants, wheezing and recurrent wheezing have been used as measures of breathing difficulties in infants. A meta-analysis of European, Latin American and African research reported prevalence rates of 36.1% (95% CI 35.2% to 37.0%) and 17.4% (95% CI 16.7% to 18.1%) for wheezing and recurring wheezing respectively for children under 2 years of age.[8] These prevalence rates were higher than rates of current wheeze in the ISAAC study for 6–7 years (11.5%) and 13–14 years (14.1%).[9] Boys have an increased likelihood of parent-reported asthma and wheeze[10–12] and receive more antiasthmatic prescriptions than girls across Denmark, Sweden and Germany.[13–15]

Most of the literature on childhood asthma and respiratory conditions does not differentiate between children with and without congenital anomalies, nor does it explore the additional burden of having chronic and frequent breathing difficulties in children with congenital anomalies. A cohort study based on health records of 170 960 Canadian children up to age six, 1980–1990, reported that children with congenital anomalies of the circulatory system and anomalies of the respiratory system had an increased risk of a recorded asthma diagnosis (HR 1.16, 95% CI 1.03 to 1.30; and HR 1.43, 95% CI 1.27 to 1.61) compared with control children.[16] Children under two with diaphragmatic hernia had high hospital readmission rates for wheeze, particularly if born preterm.[17] Most research exploring asthma and breathing difficulties in those with congenital anomalies is based on case series involving small sample sizes. For example, a cohort of adult Dutch patients with unrepaired atrial septal defect reported that four out of 31 (13%) patients had an asthma diagnosis and were taking antiasthmatic medication.[18] Similarly, a study of 14 Dutch children aged 7–12 years with oesophageal atresia identified that four of the 14 children (29%) had received a diagnosis of asthma.[19]

The aim of this study is to compare the risk of being prescribed or dispensed medications for respiratory conditions in children with and without congenital anomalies in six European regions.

## MATERIALS AND METHODS

This is a EUROlinkCAT population-based data linkage cohort study involving children from 0 to 9 years of age with and without congenital anomalies in six European regions: Denmark: Funen; Finland; Italy: Emilia Romagna; Italy: Tuscany; Spain: Valencian Region; UK: Wales.

The inclusion criteria were all liveborn children in the registry areas born between 2000 and 2014, or the first birth year included in the study by each registry (table 1). Five registries included reference children from the whole population covering the registry area, while Tuscany provided a 10% sample of reference children matched on year of birth and sex. Data on children with congenital anomalies were extracted from the European congenital anomaly (EUROCAT) registries and data on reference children were obtained from the birth registers in each region.[20–22]

Children were linked to regional or national prescription databases and to hospital outpatient pharmacy databases, if available. Children who could not be linked to these databases were classified as 'not linked' and were not included in the study, see online supplemental table 1. Wales had information on prescriptions issued by the general practitioner (GP) for approximately 70% of Welsh GP practices that were part of the Secure Anonymised Information Linkage (SAIL) databank during the study period, while the other registries had information on prescriptions dispensed by a pharmacy in their region. None of the registries were able to link to hospital inpatient prescribing data or had access to over-the-counter medication.

Antiasthmatic medication is defined using the WHO's Anatomical Therapeutic Chemical (ATC) classification system, that is, ATC codes beginning with R03 (drugs for obstructive airway diseases): beta-2 agonists (R03AC); inhaled corticosteroids (R03BA); anticholinergics (R03BB); antiallergic agents (R03BC); and leukotriene receptor antagonists (R03DC). Data on beta-2 agonists and inhaled corticosteroids were explored individually. Due to small numbers, data on the remaining antiasthmatic types were included in the 'any antiasthmatic (R03)' category. Children were classed as exposed to antiasthmatics if they had been prescribed or dispensed >1 antiasthmatic prescription in a year in order to minimise children presenting with one-time occurrences of wheeze or infection. Children were classified as unexposed if they were prescribed or dispensed less than two antiasthmatic prescriptions in a year. Data on antiasthmatic medications were included from the year 2000 or the first birth year included in the study by each registry up to the end of 2015, resulting in at least 1 year of follow-up information for each child.

Information on prescriptions was standardised according to a common data model developed as part of the EUROlinkCAT study protocol.[21 22] The EUROCAT data on children with congenital anomalies were already standardised according to EUROCAT guidelines.[23] A common analysis script was sent to all registries to produce aggregate data and analytical results. The aggregate data and analytical results were uploaded to a secure web portal for download by the research team; all individual case data were kept locally.

We explored antiasthmatic medications in children with any major congenital anomaly (defined as a congenital

**Table 1** Number and percentages of children with congenital anomalies and reference children prescribed >1 antiasthmatic medication, by registry

| Registry | Total no of children included in study | | Total receiving >1 antiasthmatic during whole follow-up | | Total receiving >1 inhaled corticosteroid during whole follow-up | | Total receiving >1 beta-2 agonist during whole follow-up | |
|---|---|---|---|---|---|---|---|---|
| | Children with congenital anomalies | Reference children | Children with congenital anomalies (%) | Reference children (%) | Children with congenital anomalies (%) | Reference children (%) | Children with congenital anomalies (%) | Reference children (%) |
| Denmark: Funen (2000–2014) | 1789 | 72290 | 614 (34.3%) | 20728 (28.7%) | 288 (16.1%) | 8114 (11.2%) | 284 (15.9%) | 7684 (10.6%) |
| Finland (2000–2014) | 32926 | 755923 | 5617 (17.1%) | 99780 (13.2%) | 3036 (9.2%) | 48866 (6.5%) | 3375 (10.3%) | 53395 (7.1%) |
| Italy: Emilia Romagna (2008–2014) | 5499 | 250829 | 2741 (49.8%) | 117675 (46.9%) | 1769 (32.2%) | 71088 (28.3%) | 1087 (19.8%) | 41215 (16.4%) |
| Italy: Tuscany (2008–2014) | 3048 | 16844 | 1392 (45.7%) | 6542 (38.8%) | 962 (31.6%) | 3947 (23.4%) | 560 (18.4%) | 1861 (11.0%) |
| Spain: Valencian Region (2010–2014) | 4281 | 223760 | 1816 (42.4%) | 77028 (34.4%) | 352 (8.2%) | 8692 (3.9%) | 740 (17.3%) | 25518 (11.4%) |
| UK: Wales (2000–2014) | 13119 | 403266 | 2540 (19.4%) | 60230 (14.9%) | 1246 (9.5%) | 26732 (6.6%) | 2018 (15.4%) | 46934 (11.6%) |
| Total | 60662 | 1722912 | 14720 (24.3%) | 381983 (22.2%) | 7653 (12.6%) | 167439 (9.7%) | 8064 (13.3%) | 176607 (10.3%) |

malformation, deformation, disruption or dysplasia),[20 21] and in 32 isolated congenital anomaly subgroups with a live birth prevalence of ≥1.75 per 10 000 births to avoid potential issues with small numbers[21] (online supplemental table 2). Isolated anomalies are defined as anomalies within a single organ, as defined using the EUROCAT algorithm.[23]

Each participating registry obtained local approvals for their data to be used in the EUROlinkCAT project.

### Patient and public involvement
No patient involved.

### Statistical analysis
Person-year estimates were calculated, which considers the number of children in the study and the length of follow-up time in the study, that is, each age group includes the number of children alive at the start of that age group. The average years of follow-up for each registry are shown in online supplemental table 1. Age groups were collapsed into six categories to avoid disclosive counts: <1 year; 1 year; 2–3 years; 4–5 years; 6–7 years; 8–9 years. Data were available from Valencian Region for children 0–5 years of age (ie, up to the children's 6th birthday), from Italian registries for children 0–7 years of age (ie, up to 8th birthday), and from Funen, Finland and Wales for children 0–9 years of age (ie, up to 10th birthday). Analyses of specific congenital anomalies were conducted using data for children aged 0–7 years to include data from the Italian registries.

DerSimonian-Laird (DL) random-effects meta-analyses were conducted to identify both the relative risk (RR) of receiving antiasthmatic prescriptions relative to the reference group, and the heterogeneity of prevalence rates across registries and by age group.[24] Random-effects meta-analyses were used to estimate RR ratios for the 32 congenital anomaly subgroups, all registries combined, and by class of antiasthmatic, that is, any antiasthmatic, inhaled corticosteroids and beta-2 agonists. Heterogeneity was estimated by the statistic $I^2$. Stata V.17 was used for data analysis.[25]

We explored the effect of receiving >1 antiasthmatic prescription by birth cohort, sex of child and gestational age (GA) separately for children with congenital anomalies and reference children, for example, children with congenital anomalies born in 2005–2009 were compared with children with congenital anomalies born 2000–2004; and similarly, reference children born in 2005–2009 were compared with reference children born 2000–2004. Birth cohort was examined for 2000–2004 (baseline group) and 2005–2009, using data from Funen, Finland and Wales as these were the only regions that had data on children born 2000–2004. For the remaining analyses, risk factors were examined for the 0–5 age group for all regions. Girls were used as the baseline group for sex of child analyses. Birth at 37+ weeks GA was used as the baseline group in comparison to birth at <32 weeks GA and birth at 32–36 weeks GA.

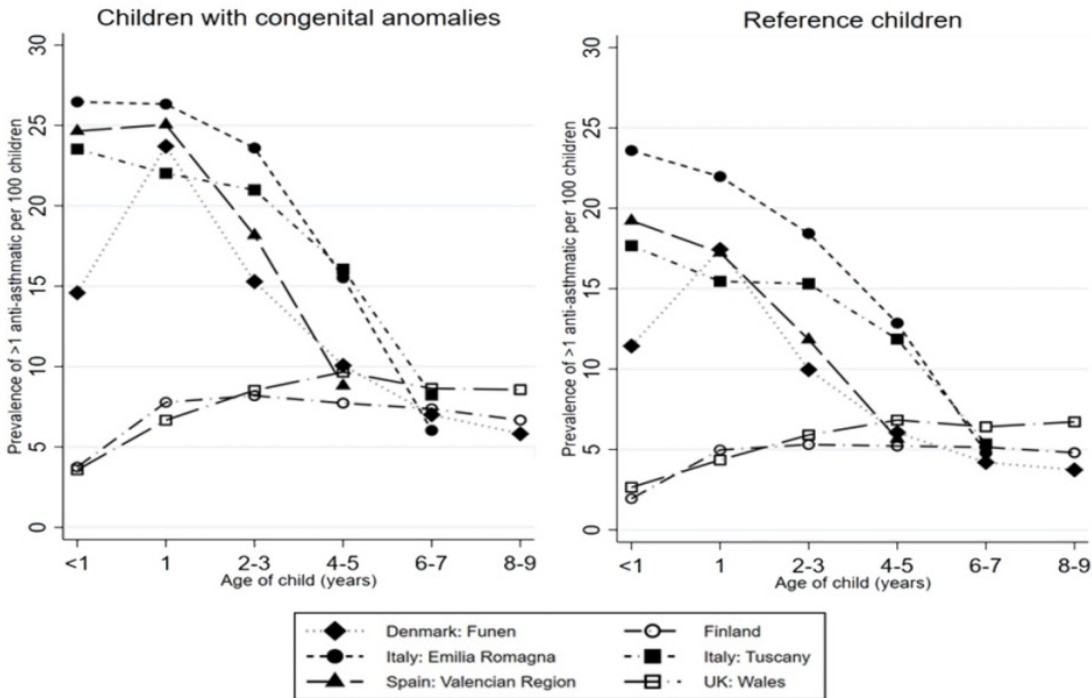

**Figure 1** Prevalence of >1 prescription for any antiasthmatic per 100 children in each age group, by registry.

Classifying children as exposed to antiasthmatics if they had been prescribed or dispensed >1 antiasthmatic prescription in a year was relaxed in a sensitivity analysis to include children with only 1 prescription/dispensation in a year.

## RESULTS

A total of 63 911 children with congenital anomalies and 1 811 431 reference children up to and including 9 years of age were included from 6 national/regional databases in 5 countries. Of these, 60 662 (94.9%) children with congenital anomalies and 1 722 912 reference children (95.1%) were included in the study. The percentage of children with and without congenital anomalies prescribed antiasthmatic medication in each region are shown in table 1. Overall, 5% of children (ranging from 0% in Tuscany and Valencian Region to 14.6% in Wales) were not included in the study as they could not be linked (online supplemental Table 1).

### Prescriptions for any antiasthmatic medicines

There were considerable geographical variations in age-specific prevalence rates of any antiasthmatic prescription (figure 1). However, the pattern of prescriptions according to age was similar for children with and without congenital anomalies in all registries. In four registries, the prevalence of >1 prescription for any antiasthmatic in children with congenital anomalies and reference children peaked at the younger ages (<1 and 1 year), then decreased sharply with age. In Finland and Wales, the prevalence increased until age 2–3 years for all children and then stabilised. Geographical differences in prevalence of >1 prescription for any antiasthmatic levelled off

by age 6–7 years, with rates in children with congenital anomalies ranging between 6% and 9% in all registries, and rates for reference children ranging between 4% and 6% in all registries. Prevalence rates for >1 prescription for any antiasthmatics were consistently slightly higher in children with congenital anomalies than reference children across registries and age groups (figure 1).

Children with congenital anomalies had a 41% higher risk of >1 prescription for any antiasthmatic compared with reference children across all age groups and across all registries (RR 1.41, 95% CI 1.35 to 1.48) (figure 2). Children aged <1 year with congenital anomalies in Finland had an almost twofold increase in risk compared with reference children (RR 1.97; 95% CI 1.86 to 2.08). The wider CIs in older age groups in some registries indicate smaller sample sizes as there were fewer children with long follow-ups. Statistically significant heterogeneity was identified between registries at all ages ($I^2$=94.3%, p<0.001) due to the large number of reference children, but the differences were small in magnitude.

### Prescriptions for inhaled corticosteroids

Children with congenital anomalies had a higher risk of receiving >1 prescription for inhaled corticosteroids than reference children. However, there was heterogeneity between age groups (p=0.004). The risk was highest for children with congenital anomalies aged <1 year (RR 2.12, 95% CI 1.50 to 2.99) and steadily decreased to 1.35 (95% CI 1.22 to 1.49) in children aged 8–9 years (figure 3). The RRs of receiving >1 prescription for inhaled corticosteroids appeared consistently higher in Valencian Region in children <1 year of age (RR 3.24, 95% CI 2.74 to 3.84), children at 1 year of age (RR 2.67, 95% CI 2.30 to 3.10)

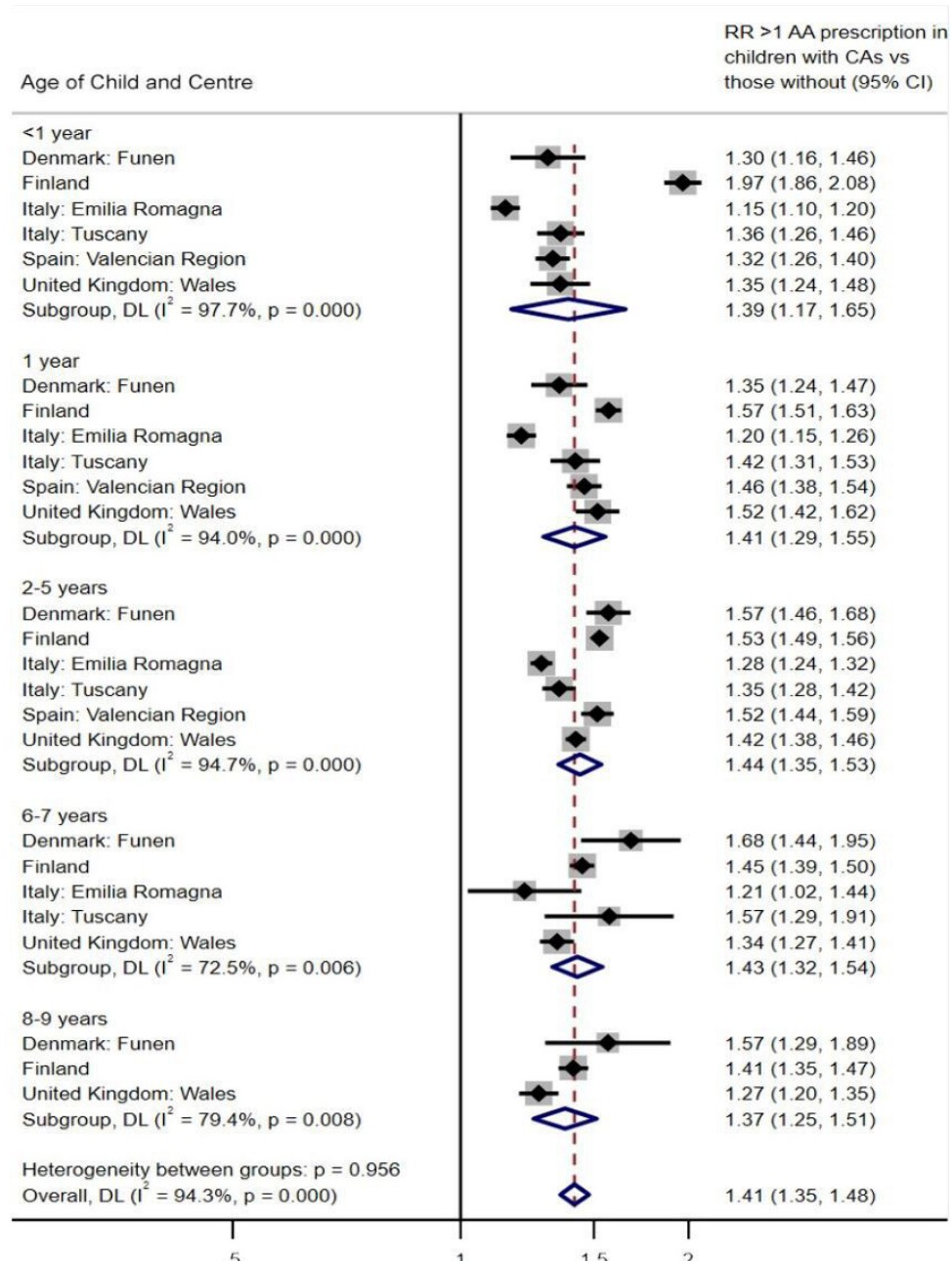

**Figure 2** Relative risk (RR) of >1 prescription for any antiasthmatic (AA) in children with congenital anomalies (CAs) compared with reference children, by registry. DL, DerSimonian-Laird.

and children at 2–5 years (RR 2.16, 95% CI 1.91 to 2.45) compared with children in other registries. Again, there was significant statistical heterogeneity across registries for all ages ($I^2$=95.6%, p<0.001) due to the large sample sizes.

### Prescriptions for beta-2 agonists

The RR of receiving >1 prescription for beta-2 agonists was 71% higher in children with congenital anomalies compared with reference children (RR 1.71, 95% CI 1.60 to 1.83) (figure 3). In contrast to inhaled cortico-steroids, the risk of receiving >1 prescription for beta-2-agonists remained consistently high across all age groups (heterogeneity between age groups p=0.358). Significant

heterogeneity was identified across registries for all ages ($I^2$=94.4%, p<0.001).

### Risk factors for receiving antiasthmatic prescriptions (children up to age 6 years)

Children aged 0–5 years were more likely to receive >1 antiasthmatic per year in 2005–2009 compared with 2000–2004, with this risk being higher in reference children than children with congenital anomalies (table 2). Children with congenital anomalies born at <32 weeks GA had over twice the risk of being prescribed/dispensed antiasthmatics (RR 2.20, 95% CI 2.10 to 2.30) compared with children born at 37+ weeks. The risk was also raised for reference children (RR 1.86, 95% CI 1.82 to 1.90).

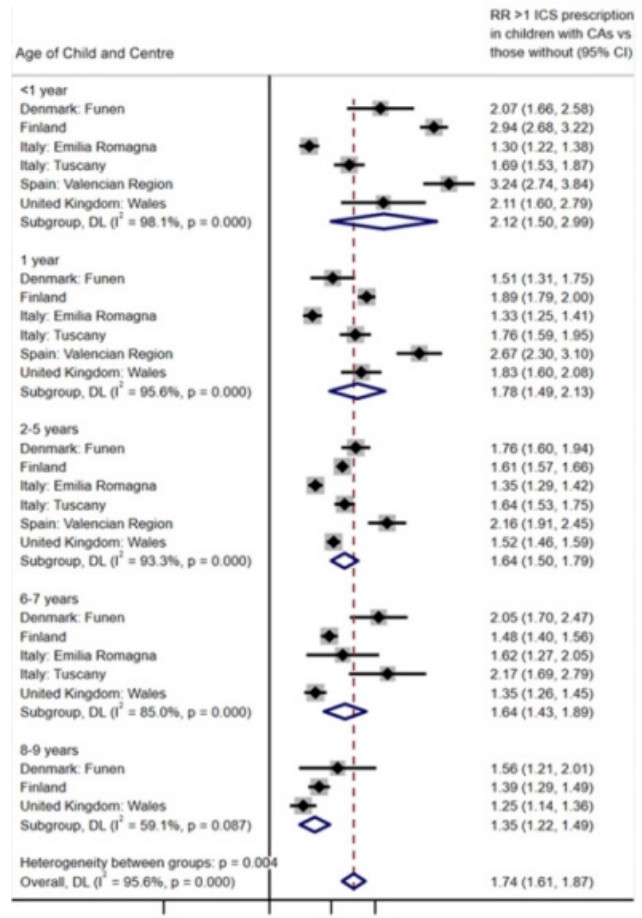

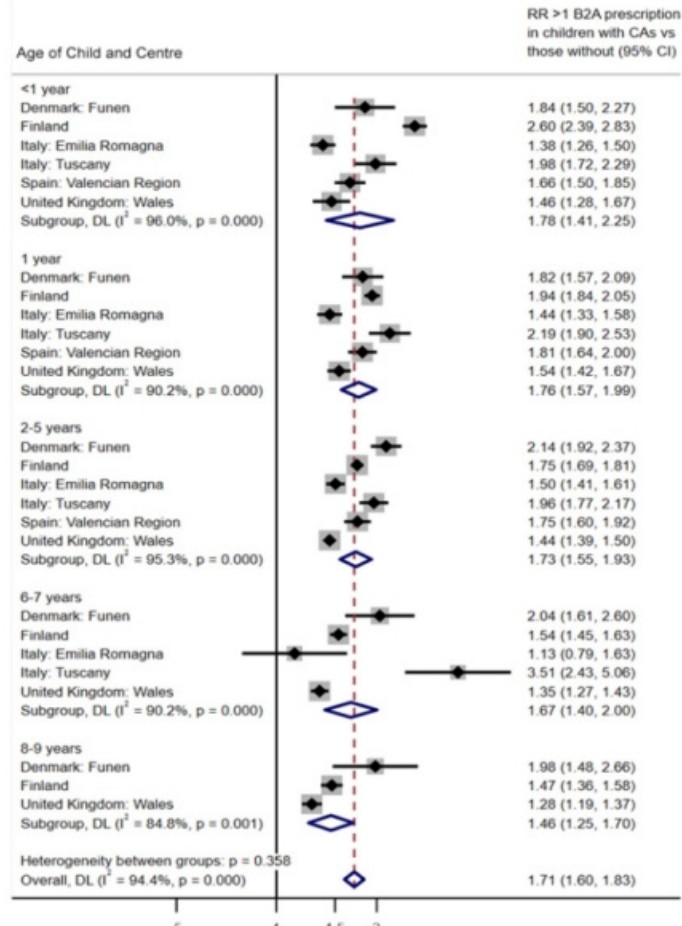

**Figure 3** Relative risk (RR) of >1 beta-2 agonist (B2A) prescription and >1 inhaled corticosteroid (ICS) prescription in children with congenital anomalies (CAs) compared with reference children, by registry. DL, DerSimonian-Laird.

RRs decreased in children born 32–36 weeks GA but remained higher in children with congenital anomalies than in reference children (RR 1.45, 95% CI 1.40 to 1.49; and RR 1.31 95% CI 1.30 to 1.32, respectively). Boys were more likely to be prescribed antiasthmatics than girls, with similar risks observed for children with congenital anomalies and reference children.

**Congenital anomaly subgroups (children up to 8 years of age)**

The RR for having >1 antiasthmatic prescription for children with specific congenital anomalies aged 0–7 years was compared with reference children. Children with oesophageal atresia had the highest risk of being prescribed/dispensed any antiasthmatic medication (RR 3.57, 95% CI 3.14 to 4.06), inhaled corticosteroids (RR 5.46, 95% CI 4.46 to 6.68) and beta-2 agonists (RR 5.75, 95% CI 4.86 to 6.81) (online supplemental table 3). Children with genetic syndromes (Di George syndrome and Noonan syndrome), chromosomal anomalies (all chromosomal anomalies and Down syndrome with and without congenital heart defects), and diaphragmatic hernia had more than double the risk of having >1 prescription for any

**Table 2** Relative risk (RR) of >1 antiasthmatic prescriptions in children with congenital anomalies and reference children up to age 6 years, by risk factor for the birth cohort analysis, data were analysed only from three registries: Denmark: Funen, Finland and Wales

|  | Children with congenital anomalies (RR; 95% CI) | Reference children (RR; 95% CI) |
|---|---|---|
| Birth cohort | | |
| 2000–2004 | Reference | |
| 2005–2009 | 1.27 (1.23 to 1.31) | 1.60 (1.59 to 1.61) |
| Gestation age | | |
| 37+ weeks | Reference | |
| 32–36 weeks | 1.45 (1.40 to 1.49) | 1.31 (1.30 to 1.32) |
| <32 weeks | 2.20 (2.10 to 2.30) | 1.86 (1.83 to 1.90) |
| Sex of child | | |
| Female | Reference | |
| Male | 1.45 (1.40 to 1.49) | 1.31 (1.30 to 1.32) |

antiasthmatics compared with reference children. Across all 32 congenital anomaly subgroups examined, children with these anomalies were significantly more likely to be prescribed any antiasthmatic than reference children, except for children with tetralogy of Fallot or children with hip dislocation and/or dysplasia whose risks were slightly raised but were not statistically significant.

The RR for receiving >1 prescription for beta-2-agonists and for inhaled corticosteroids for the 32 congenital anomaly subgroups followed the same trends as those for all antiasthmatic medication, but they were less likely to be statistically significant due to smaller numbers being analysed.

### Sensitivity analysis

The overall RR of children with congenital anomalies receiving at least one prescription for any antiasthmatic medication was 1.25 (95% CI 1.21 to 1.28) compared with reference children (online supplemental table 4). This is compared with the RR=1.41 (95% CI 1.35 to 1.48) of receiving >1 prescription. Although the risk estimates were lower for children with at least one prescription for any antiasthmatic, the pattern of risk across registries and age groups was similar to that found for children with >1 prescription (online supplemental figure 1).

### DISCUSSION

This population-based cohort study of over 1.78 million children in six European regions found that across all regions and age groups, children with congenital anomalies had a consistently higher risk of receiving prescriptions for >1 antiasthmatic medication compared with reference children. We observed geographical variation in the use of antiasthmatic medications with registries in Denmark, Italy and Spain having higher prevalence in young children whereas Finland and Wales had lower prevalence. By age 6–7 years, the prevalence in the individual registries started to converge for both children with and without anomalies, although it remained slightly higher for children with congenital anomalies. Children with oesophageal atresia, diaphragmatic hernia, chromosomal anomalies and specific genetic syndromes had the highest risk of being prescribed antiasthmatic prescriptions. Male children, children born in a later cohort, and children born preterm had an increased risk of antiasthmatic prescriptions, with children with congenital anomalies born very preterm (<32 weeks) having over double the risk of antiasthmatic prescriptions compared with children with congenital anomalies born at term.

With the exception of Finland and Wales, our findings that antiasthmatic prescriptions tended to be higher in younger age groups is consistent with literature indicating higher rates of wheezing and current wheeze in infants below 2 years of age than older children.[8][9] As beta-2 agonists and inhaled corticosteroids are recommended as treatment for infant wheeze,[26] prescriptions for antiasthmatics in young children may be indicative of wheeze and acute respiratory infections which can be common in young children.[27] However, other studies reported higher prevalence of breathing difficulties for older children in the form of parent-reported asthma.[3] It is possible that parental reports may overestimate asthma prevalence, while our study was based on administrative antiasthmatic prescription records. We identified antiasthmatic prevalence rates that seemed to vary from other regional studies. For example, a study based on a sample of Welsh children (n=1529), born 2000–2002, with a GP diagnosis of asthma and/or prescriptions for antiasthmatics in their health records in the last 12 months reported much higher prevalence rates ranging from 13.4% for 3 years to 10.4% in 7 years.[28] These differences may relate to methodology as our study criteria was >1 antiasthmatic prescription in a year rather than at least one. We also identified regional differences within countries as within Italy, children in Emilia Romagna had a higher prevalence of >1 prescription for asthma medications than children in Tuscany. A study investigating the use of antiasthmatics in women before, during and after pregnancy also reported this regional difference between the two Italian registries.[29]

Children with congenital anomalies consistently had higher rates of antiasthmatic medication than reference children which may be due to the fact that antiasthmatic medications are widely used to treat respiratory complications of some anomalies. For example, congenital anomalies such as diaphragmatic hernia and Down syndrome are associated with pulmonary hypoplasia,[30–33] requiring antiasthmatics to manage wheezing episodes. Children with diaphragmatic hernia commonly require mechanical ventilation for days or weeks after birth which can lead to bronchopulmonary dysplasia.[31][34] Beta-2 agonists are prescribed to children with bronchopulmonary dysplasia to reduce pulmonary resistance.[35] Studies have shown that patients with diaphragmatic hernia were over six times more likely to have been diagnosed with asthma than the general population.[34] Congenital anomalies may be comorbid with respiratory conditions, such as the associations between oesophageal atresia, Down syndrome and Di George syndrome with respiratory tract disease.[36–38] In addition, children with congenital anomalies requiring surgery and follow-up in the first year of life or those requiring routine medical care throughout their life span will have multiple interactions with the medical care team, hence there is greater opportunity for a diagnosis of respiratory difficulties to be made and antiasthmatics to be dispensed. Indeed, an earlier EUROlinkCAT study found that children with congenital anomalies had significantly more hospital stays than children without congenital anomalies, particularly in the first year of life.[39]

We did not find an increased risk for prescription of beta-2 agonists for children with tetralogy of Fallot. Beta-2-agonists should not be used in children with infundibular pulmonary stenosis as beta-blockers may be prescribed to prevent hypoxic spells in these children while awaiting surgery.[40] While over 50% of children in

hospitals are administered 'unlicensed' or 'off-label' medicines,[41 42] a Spanish study reported that 16% of anti-asthmatics dispensed by community pharmacies for use by children were 'off-label' and that the seasonal variations suggested that these were prescribed to treat respiratory infections in young children rather than asthma.[43] We do not have information about off-label use in our study but the indications and age limits for prescribing off-label antiasthmatics are likely to differ between countries.

The finding that boys were more likely to be prescribed/dispensed antiasthmatics is consistent with previous literature.[10–15] Both children with and without congenital anomalies had more antiasthmatic prescriptions in 2005–2009 compared with 2000–2004. While the magnitude of risk was higher in reference children than for children with congenital anomalies, the comparisons were not between children with congenital anomalies and reference children, but within the two groups of children. Indeed, children with congenital anomalies had a slightly higher proportion of >1 antiasthmatic prescription (24%) compared with reference children (22%). While self-reported antiasthmatic use has increased in cohort studies of young adults between 1990 and 2007,[44] our findings support an increasing trend of antiasthmatic prescriptions over time in children using administrative prescription data. Children born preterm at <32 weeks GA were twice as likely to be prescribed/dispensed antiasthmatics than children born at term, although the risk was reduced for children born 32–36 weeks. The higher risk in children born <32 weeks GA may be due to respiratory complications such as respiratory distress syndrome or bronchopulmonary dysplasia arising from mechanical ventilation in preterm children[45 46]

A major strength of this study is that we have standardised data on congenital anomalies and a common data model which enabled us to map and standardise the prescription/pharmacy dispensing records, as well as standardising demographic information on children. This is important as there are diverse coding classification systems, languages and healthcare systems in Europe. Furthermore, this is the first population-based multicentre study exploring antiasthmatic prescriptions in children with and without congenital anomalies which included data from 60 662 children with congenital anomalies across Europe, overcoming limitations of previous research based on small sizes. The use of reference children for comparison in each geographical region is also a major strength, as some differences in prescription rates across Europe may be explained by methodological issues (such as frequency of prescriptions issued at the pharmacy) and different level of indication for prescribing asthma medications to small children.

However, there were a number of limitations to this study. First, bias may arise as approximately 5% of children across registries were unable to be linked to the prescription database due to invalid identifications numbers as was the case in Tuscany, or because their records were not sent to the national/vital statistics to be linked. In Wales,

the children were registered in a GP practice that was not part of the SAIL databank, therefore, they could not be linked. There is evidence suggesting that low socioeconomic status (SES) is associated with asthma.[47] Wales compared the SES of children registered in SAIL vs children not registered in SAIL and concluded that there were no differences in SES between linked/unlinked children. Similarly, Tuscany found no differences in GA, maternal age, sex and survival between children with congenital anomalies who were linked versus those who were not linked. Second, only three participating registries (Funen, Finland and Wales) could provide data for children up to and including 9 years of age. Our list of antiasthmatic medications did not include oral corticosteroids used to treat acute symptoms, but for young children this treatment may mainly be done during hospital stays, and we were unable to differentiate between long-acting and short-acting beta-2-agonists. While we explored medications prescribed and dispensed, we were unable to gauge whether these medications were actually taken.

In summary, this study demonstrates that valid information on the prevalence of antiasthmatic prescriptions for children can be obtained through data linkage studies in order to monitor regional differences across age groups and over time. In general, children with congenital anomalies had a higher prevalence and risk of antiasthmatic prescriptions than reference children. To evaluate the additional burden of disease that children with congenital anomalies face, it is essential to compare these children to those without congenital anomalies in the same geographical areas as prescription patterns of antiasthmatic medications vary across Europe. The findings from this study are also beneficial to clinicians for identifying which congenital anomalies are associated with a higher risk of prescriptions for respiratory symptoms and breathing difficulties.

**Author affiliations**
[1]Institute of Nursing and Health Research, Faculty of Life and Health Sciences,Ulster University, Belfast, UK
[2]Population Health Research Institute, St George's University of London, London, UK
[3]Emilia Romagna Registry of Birth Defects, University of Ferrara, Ferrara, Italy
[4]Department of Neuroscience and Rehabilitation, University of Ferrara, Ferrara, Italy
[5]Rare Diseases Research Unit, Foundation for the Promotion of Health and Biomedical Research in the Valencian region, Valencia, Spain
[6]Institute of Clinical Physiology, National Research Council Pisa Research Area, Pisa, Italy
[7]Department of Paediatrics and Adolescent Medicine, Lillebaelt Hospital, University Hospital of Southern Denmark, Kolding, Denmark
[8]Department of Knowledge Brokers, THL Finnish Institute for Health and Welfare, Helsinki, Finland
[9]Department of Nursing, Swansea University, Swansea, UK
[10]Institute of Clinical Physiology National Research Council, Pisa, Italy
[11]Section of Epidemiology, Department of Public Health, University of Copenhagen, Copenhagen, Denmark

**Contributors** ND prepared the manuscript and revised it following feedback and comments from all authors. ND analysed the pooled data from each registry. JEG and ML produced the standardisation scripts. JT and JKM produced the central analysis scripts. GA, EB, LB-B, CC-C, AC, MG, AH, SJ, AP, IS and SKU extracted and analysed local registry data. ML, EG and JKM secured funding, conceptualised and designed the study, and provided frequent feedback on manuscript drafts. JKM and

ML provided data analysis guidance and syntax for meta-analyses. ML executed data analysis syntax relating to protected data. ML revised the manuscript following comments from reviewers. ML is the study guarantor.

**Funding** This work was supported by the European Union's Horizon 2020 research and innovation programme under grant agreement No. 733001.

**Disclaimer** The views presented here are those of the authors only, and the European Commission is not responsible for any use that may be made of the information presented here.

**Competing interests** None declared.

**Patient and public involvement** Patients and/or the public were not involved in the design, or conduct, or reporting, or dissemination plans of this research.

**Patient consent for publication** Not applicable.

**Ethics approval** Ethical approval for this study was obtained from Ulster University's Nursing and Health Research Ethics Filter Committee (Reference Number: FCNUR-21-060). This study is based on secondary data analysis of data held in routine administrative databases in Europe. The individual records of children linked to electronic prescription databases remain within the regional/national statistical organisations. Only aggregated results were sent to the research team for analysis, hence informed consent was not required. Each participating registry obtained local approvals for their data to be used in the EUROlinkCAT project as described in the paper published in BMJ Open: Claridge H et al. Ethics and legal requirements for data linkage in 14 European countries for children with congenital anomalies. BMJ Open 2023; 13:e071687. doi: 10.1136/bmjopen-2023-071687 https://bmjopen.bmj.com/content/13/7/e071687

**Provenance and peer review** Not commissioned; externally peer reviewed.

**Data availability statement** Data may be obtained from a third party and are not publicly available. We are legally not allowed to share the third-party administrative data used in this study as it belongs to the data providers in each of the regions ie, the regional or national statistical organisations. All our documentation is available on the EUROlinkCAT website (http://www.EUROlinkCAT.eu/wp2-buildingresultsr epository) and we encourage any interested parties to apply to the EUROlinkCAT management team to assist them in obtaining approval from the data providers in each region/country to use the aggregated data for an approved study http://www. EUROlinkCAT.eu/contactinformationanddatarequests.

**ORCID iDs**
Natalie Divin http://orcid.org/0000-0002-6063-2451
Joanne Emma Given http://orcid.org/0000-0003-4921-1944
Clara Cavero-Carbonell http://orcid.org/0000-0002-4858-6456
Ester Garne http://orcid.org/0000-0003-0430-2594
Susan Jordan http://orcid.org/0000-0002-5691-2987
Joan K Morris http://orcid.org/0000-0002-7164-612X
Maria Loane http://orcid.org/0000-0002-1206-3637

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
