## [Reviewer comments · BMJ Open]

ARTICLE DETAILS

TITLE (PROVISIONAL)	Anti-asthmatic prescriptions in children with and without congenital anomalies: a population-based study
AUTHORS	Divin, Natalie; Given, Joanne; Tan, Joachim; Astolfi, Gianni; Ballardini, Elisa; Barrachina-Bonet, Laia; Caverro- Carbonell, Clara; Coi, Alessio; Garne, Ester; Gissler, Mika; Heino, Anna; Jordan, Susan; Pierini, Anna; Scanlon, Ieuan; Urhøj, Stine Kjær; Morris, Joan; Loane , Maria

VERSION 1 – REVIEW

REVIEWER	Crossingham, Ian East Lancashire Hospitals NHS Trust I work clinically in adult (not paediatric) respiratory medicine. I have no other declarations of interest.
REVIEW RETURNED	28-Dec-2022

GENERAL COMMENTS	Many thanks for sending me this manuscript for review. It describes a Europe wide cohort study of respiratory prescribing in children drawn from existing databases. There can be a distinct feeling that such cohorts drawn from multiple sources are a bit "stitched together" but the inevitable heterogeneity has been properly explored here. The central finding - that children with congenital anomalies have an increased rate of prescriptions for obstructive airways disease - is an interesting one and clearly warrants further investigation. Are there mechanistic links between such anomalies and asthma? Are physicians using asthma medications off licence to treat respiratory complications of the anomalies? Does having a congenital anomaly lead to greater exposure to doctors and thus more opportunity for an asthma diagnosis to be made? The discussion commendably avoids speculating. The writing is clear and concise. Any quibbles I may have are minor. Suggest push forward with publication.
---

REVIEWER	Howley, Meredith Birth Defects Research Section, New York State Department of Health
REVIEW RETURNED	15-Mar-2023

GENERAL COMMENTS	The authors sought to compare the risk of being prescribed an anti-asthmatic medication in children with congenital anomalies and those without a congenital anomaly. To do so, they pooled data in six European regions and relied on birth defect registry information
--

	and prescription medication registry data. This was a well-done analysis that relied on existing data and data linkages to explore the question and the manuscript was clearly written. I had a few comments for the authors to consider:  1. The authors discuss that bias may arise from the inability to link to prescription database (page 12, line 327). The percentage of participants from Wales that were unlinked was much higher than the percentages from the other regions and the authors suggested this was due to children in GP practices that were not part of the SAIL database. Is there any reason to think that those registered in a GP practice that was not part of SAIL databank would be more likely to take an anti-asthmatic medication? Are there differences (in age, region within Wales, socioeconomic differences) in who is in the SAIL database and who is not? I think it might be worth describing the impact of this potential bias on the results, more than just acknowledging that it exists as I was struggling to make sense of how this might have impacted the results. Similarly, 11.5% of children with anomalies in Italy, Tuscany could not be linked, while 0 reference children from that area could not be linked. Do the authors have thoughts on that difference? 2. I was curious how the authors accounted for deaths within the analysis. To have been prescribed an anti-asthmatic at an older age, the child had to have survived to that age which is less likely for those with a birth defect. Was information on deaths available from the birth defect registries and accounted for in any way? Do we know that all children in the analysis of each age group were alive at each age to have had the chance to be prescribed a medication? If not, I think this should be included in the discussion. 3. In Table 2, the risks by birth cohort were higher in reference children (1.60) than in children within anomalies (1.27). While I realize that both those with anomalies and those without had increased risk of asthma medication prescription in 2005-2009 compared to 2000-2004, why do the authors make of the higher risk among the reference group? The discussion does not explain this finding or offer clues to why this might be, which I think would be helpful as it is opposite of the rest of the findings. believe that the reference children had higher risk?
--	--

VERSION 1 – AUTHOR RESPONSE

Reviewer 2: The authors discuss that bias may arise from the inability to link to prescription database (page 12, line 327). The percentage of participants from Wales that were unlinked was much higher than the percentages from the other regions and the authors suggested this was due to children in GP practices that were not part of the SAIL database. Is there any reason to think that those registered in a GP practice that was not part of SAIL databank would	We investigated this issue in Wales and Tuscany: Wales: By the end of 2022, 86.5% of children in Wales were included in the Wales Longitudinal General Practice (WLGP) dataset, with information on prescription medicines. In this study, the population with prescription data had a similar mean deprivation (Townsend) score [1] to those without prescription data (0.603 [3.678] vs 0.612 [3.580]), which was not statistically significant. We conclude that the Welsh prescription data in this study are based on a sample of children that is representative of children in Wales in terms of socio-economic status.  1. Townsend, P. (1987) Deprivation. Journal of Social Policy, 16, 125-146. http://dx.doi.org/10.1017/S0047279400020341
---	---

be more likely to take an anti-asthmatic medication? Are there differences (in age, region within Wales, socioeconomic differences) in who is in the SAIL database and who is not? I think it might be worth describing the impact of this potential bias on the results, more than just acknowledging that it exists as I was struggling to make sense of how this might have impacted the results. Similarly, 11.5% of children with anomalies in Italy, Tuscany could not be linked, while 0 reference children from that area could not be linked. Do the authors have thoughts on that difference?	Tuscany: Tuscany was the only registry that included a 10% sample of reference children (the other 5 registries included all reference children in the registry population). The reference children were selected from the registry birth records hence they had a valid ID number that could be linked to the healthcare databases. In contrast 11.5% of children with congenital anomalies could not be linked due to issues with ID numbers, particularly within the first months of life. Investigations comparing children with congenital anomalies who were linked to those who were not linked found no significant difference on several risk factors (maternal age, gestational age, survival, year of birth, sex). We have amended the text in the Discussion, changes are in blue font: Firstly, bias may arise as approximately 5% of children across registries were unable to be linked to the prescription database due to invalid identification numbers as was the case in Tuscany, or because their records were not sent to the national/vital statistics to be linked. In Wales, the children were registered in a GP practice that was not part of the SAIL databank, therefore they could not be linked. There is evidence suggesting that low socioeconomic status (SES) is associated with asthma¹ Wales compared the SES of children registered in SAIL versus children not registered in SAIL and concluded that there were no differences in SES between linked/unlinked children. Similarly, Tuscany found no differences in gestational age, maternal age, sex and survival between children with congenital anomalies who were linked versus those who were not linked. [1. Uphoff E, Cabieses B, Pinart M, et al. A systematic review of socioeconomic position in relation to asthma and allergic diseases. Eur Respir J 2015; 46: 364–374 PubMed]
I was curious how the authors accounted for deaths within the analysis. To have been prescribed an anti-asthmatic at an older age, the child had to have survived to that age which is less likely for those with a birth defect. Was information on deaths available from the birth defect registries and accounted for in any way? Do we know that all children in the analysis of each age group were alive at each age to have had the chance to be prescribed a medication? If not, I think this should be included in the discussion.	Complete information on deaths was available. We used “Person-Year” analyses which takes into account the number of children in the study and the amount of time each child stayed in the study i.e. the follow-up time. Within each group, we included the children alive at the start of the age group. If a child died in a particular age group, that child was not included in the next consecutive age group. We have clarified this at the beginning of the Statistical analysis section of the Methods, by adding the text below: Person-year estimates were calculated, which considers the number of children in the study and the length of follow-up time each child is in the study i.e. each age group includes the number of children alive at the start of that age group.
In Table 2, the risks by birth cohort were higher in reference children (1.60) than in children within anomalies (1.27). While I realize that both those with anomalies	In table 2 the risks for reference children born in 2005-2009 were compared to reference children born 2000-2004; and similarly, children with congenital anomalies born 2005-2009 were compared to children with congenital anomalies born 2000-2004. Therefore, the RRs of 1.60 and 1.27 mean that for reference children there was a 60% increased risk of asthma medication

and those without had increased risk of asthma medication prescription in 2005-2009 compared to 2000-2004, why do the authors make of the higher risk among the reference group? The discussion does not explain this finding or offer clues to why this might be, which I think would be helpful as it is opposite of the rest of the findings. believe that the reference children had higher risk?	being prescribed, whereas in children with congenital anomalies the risk only increased by 27% over the same time period. Table 1 shows that the actual amount of prescribing is greater in children with congenital anomalies than in reference children. The greater increase in reference children may reflect changes in attitudes to prescribing which may be more apparent in less “ill” children. However, this is speculation so we have not included it in the text. We have clarified the text in the methods: We explored the effect of receiving >1 anti-asthmatic prescription by birth cohort, sex of child, and gestational age (GA) separately for children with congenital anomalies and reference children e.g. children with congenital anomalies born in 2005-2009 were compared to children with congenital anomalies born 2000-2004; and similarly, reference children born in 2005-2009 were compared to reference children born 2000-2004. We added the following sentence to the Discussion: While the magnitude of risk was higher in reference children than for children with congenital anomalies, the comparisons were not between children with congenital anomalies and reference children, but within the two groups of children. Indeed, children with congenital anomalies had a slightly higher proportion of >1 anti-asthmatic prescription (24%) compared to reference children (22%).
--	---

VERSION 2 – REVIEW

REVIEWER	Howley, Meredith Birth Defects Research Section, New York State Department of Health
REVIEW RETURNED	25-May-2023
GENERAL COMMENTS	Thank you for looking into my comments and answering my questions from the first review. I am satisfied with the response and have no other concerns.